# Identification of critical genetic variants associated with metabolic phenotypes of the Japanese population

Seizo Koshiba[1,2,3]✉, Ikuko N. Motoike[1,4], Daisuke Saigusa[1,2], Jin Inoue[1,2], Yuichi Aoki[1,4], Shu Tadaka [1,4], Matsuyuki Shirota[1,2], Fumiki Katsuoka[1,2,3], Gen Tamiya[1,2,3], Naoko Minegishi[1,2,3], Nobuo Fuse [1,2,3], Kengo Kinoshita [1,3,4] & Masayuki Yamamoto [1,2,3]✉

We performed a metabolome genome-wide association study for the Japanese population in the prospective cohort study of Tohoku Medical Megabank. By combining whole-genome sequencing and nontarget metabolome analyses, we identified a large number of novel associations between genetic variants and plasma metabolites. Of the identified metabolite-associated genes, approximately half have already been shown to be involved in various diseases. We identified metabolite-associated genes involved in the metabolism of xeno-biotics, some of which are from intestinal microorganisms, indicating that the identified genetic variants also markedly influence the interaction between the host and symbiotic bacteria. We also identified five associations that appeared to be female-specific. A number of rare variants that influence metabolite levels were also found, and combinations of common and rare variants influenced the metabolite levels more profoundly. These results support our contention that metabolic phenotyping provides important insights into how genetic and environmental factors provoke human diseases.

[1] Tohoku Medical Megabank Organization, Tohoku University, 2-1, Seiryo-machi, Aoba-ku, Sendai 980-8573, Japan. [2] Graduate School of Medicine, Tohoku University, 2-1, Seiryo-machi, Aoba-ku, Sendai 980-8575, Japan. [3] The Advanced Research Center for Innovations in Next-Generation Medicine (INGEM), Tohoku University, 2-1, Seiryo-machi, Aoba-ku, Sendai 980-8573, Japan. [4] Graduate School of Information Sciences, Tohoku University, 6-3-09, Aramaki Aza-Aoba, Aoba-ku, Sendai 980-8579, Japan. ✉email: koshiba@megabank.tohoku.ac.jp; masiyamamoto@med.tohoku.ac.jp

Molecular phenotyping by means of multiomics analyses is an indispensable approach for modern medical sciences and practices. In the last decade, large-scale population-based prospective cohort studies have applied molecular phenotyping for participants, and obtained a number of important insights into the causes of phenotypes[1–10]. Among omics analyses, metabolite profiling is one of the most powerful methods for describing individual phenotypes, as the metabolite profile is heavily influenced by both genetic and environmental factors, even if the effect size of each influencing factor is too small to be detected in apparent phenotypes. In this regard, metabolome genome-wide association studies (MGWASs) enabled us to assess the influence of common genetic variants on phenotypes in an effective and elaborate manner[1,2,4,5,9,11–16].

Although previous MGWASs identified certain number of associations, the magnitude of the analyses was not large enough to comprehensively elucidate the effects of genetic variants on the metabolic phenotypes. One major limitation inherent to conventional MGWASs is that these studies mainly use DNA array technology for genotyping of individuals, resulting in only a limited number of genetic variants being applied in the MGWAS analyses. Another limitation is that although the frequency of each genetic variant differs considerably among ethnic groups, most previous MGWASs targeted wide-ranging ethnic groups *en bloc*. Thus, there remain many unknown genetic variants influencing metabolic phenotypes, so closer investigations of the metabolite profiles of individual ethnic groups are very important for comprehensively elucidating the genetic effects on various metabolic phenotypes.

It has been generally hypothesized that the effects of rare variants on metabolic phenotypes may be stronger than those of common variants, but most of the association studies to date have not addressed this point because of the technical limitations of the DNA microarray-based GWAS system. Therefore, in this study, we tested an original approach in which we first identified genes that harbor effective common variants in an MGWAS of 1008 participants using whole-genome sequence datasets, and then searched for rare variants within the identified genes[17]. Moreover, to comprehensively elucidate the relationship between genetic variants and metabolite profiles in human plasma, we expanded the number of metabolites used for MGWAS analyses by combining nuclear magnetic resonance (NMR)-based metabolome analysis with nontarget liquid chromatography-mass spectrometry (LC-MS)-based metabolome analysis for 1008 participants.

In this study, our MGWAS analysis based on whole-genome sequencing and nontarget metabolome data revealed many novel associations between genetic variants and metabolites. We found that approximately half of the associated genes are involved in various diseases, indicating that the metabolome provides an important intermediate phenotype for investigation of disease causes. In addition, we found that metabolites involved in many kinds of xenobiotic metabolism were associated with genetic variants, and some of the xenobiotics were from intestinal bacteria, indicating that the cross-talk between host and microbes is heavily influenced by host gene polymorphisms. On the other hand, we also showed that many rare variants influenced metabolic levels more strikingly than the common variants in the corresponding genes. Importantly, various combinations of the rare and common variants frequently influenced metabolic levels more profoundly than single variants. Our results thus indicate that the MGWAS approach significantly enhances not only the functional annotation of genetic variants, but also the study of the effects of these variants on diseases.

## Results

### MGWAS identified many novel genetic loci associated with plasma metabolite levels.
A total of 1008 participants in TMM Community-Based Cohort Study were selected for the MGWAS analyses. We conducted nontarget metabolome analyses of plasma samples from these participants using both NMR spectroscopy and LC-MS. After validating the quality of the data, 37 and 270 metabolites obtained by NMR spectroscopy and LC-MS, respectively, were selected with 255 nonredundant metabolites excluding overlapping metabolites (Supplementary Data 1) and used for subsequent association studies. MGWAS analyses were performed utilizing approximately 10-million variants from the whole-genome sequence data analyzed in the TMM project[17–19]. We identified 42 significant associations of 38 plasma metabolites with 33 genetic variants (in 26 loci) at a genome-wide significant *P*-value threshold (Fig. 1a). Of these 26 associated loci, 11 loci have not been reported in previous MGWASs (Tables 1, 2 and Supplementary Data 2). The 38 associated metabolites found in this study corresponded to amino acids, lipids, fatty acids, carbohydrates, nucleic acids, and their derivatives, including those from the gut microbiota (Tables 1 and 2), indicating that the genetic variants identified in this study influenced many metabolic pathways. In addition, we also identified five associations that were significant for only female samples (Table 2). In contrast, we could not identify any associations that were significant for only male samples.

We also conducted a replication analysis based on another set of participants from the same cohort. We selected an additional 295 participants (130 female) for whom whole-genome sequence datasets were available and conducted metabolome analyses in a similar manner to the discovery study. Because the number of participants for the replication was limited, we could analyze the significant associations of variants with an allele frequency of greater than 0.05 (MAF > 0.05). As we could not include enough number of females for the replication study, associations significant only for females could not be pursued. Among 24 target associations (16 loci), 13 were replicated ($p ≤ 0.05/16 = 0.0031$), and 7 were nominally replicated ($P ≤ 0.05$) (Tables 1 and 2). Among the remaining four associations, one (*SLC7A5* with L-kynurenine) was previously reported in other MGWASs, while two (*FADSs* with two lipids) were the associations with the *FADS* locus, which is well-known to associate with a wide variety of lipids. These results show that most of the associations found in the discovery study were replicated and that the remaining ones would be replicated if the number of samples is increased.

**Metabolite associations with nonsynonymous variants**. We identified eight nonsynonymous variants, which consisted of six missense and two nonsense (i.e., stop-gain) variants (Table 1). Among these variants, five missense variants associated with five metabolites (glycine, proline, asparagine, phenylalanine, and formate) were identified previously through the analyses of approximately 500 participants[14]. Thus, doubling of the participant number and renewing the analytical methods resulted in the detection of three new loci and five new associations (Table 1). Besides our current study suitably proved the reproducibility of the MGWAS analyses[14], the increase also resulted in more significant *P*-values for all five known associations compared with the previous study (Table 1 and Supplementary Fig. 1a–e).

**Two newly identified stop-gain variants**. We detected newly two nonsense (stop-gain) variants associated with metabolites. One is rs121907892, which is in the solute carrier family 22-member-12 (*SLC22A12*) gene and results in a stop codon at residue 258 (tryptophan) of the gene (Table 1 and Supplementary Fig. 2a). The minor allele variant of this SNP decreases the plasma levels of urate (Fig. 1b). The SLC22A12 protein, called URAT1, is a urate transporter that regulates blood urate levels by reabsorption of

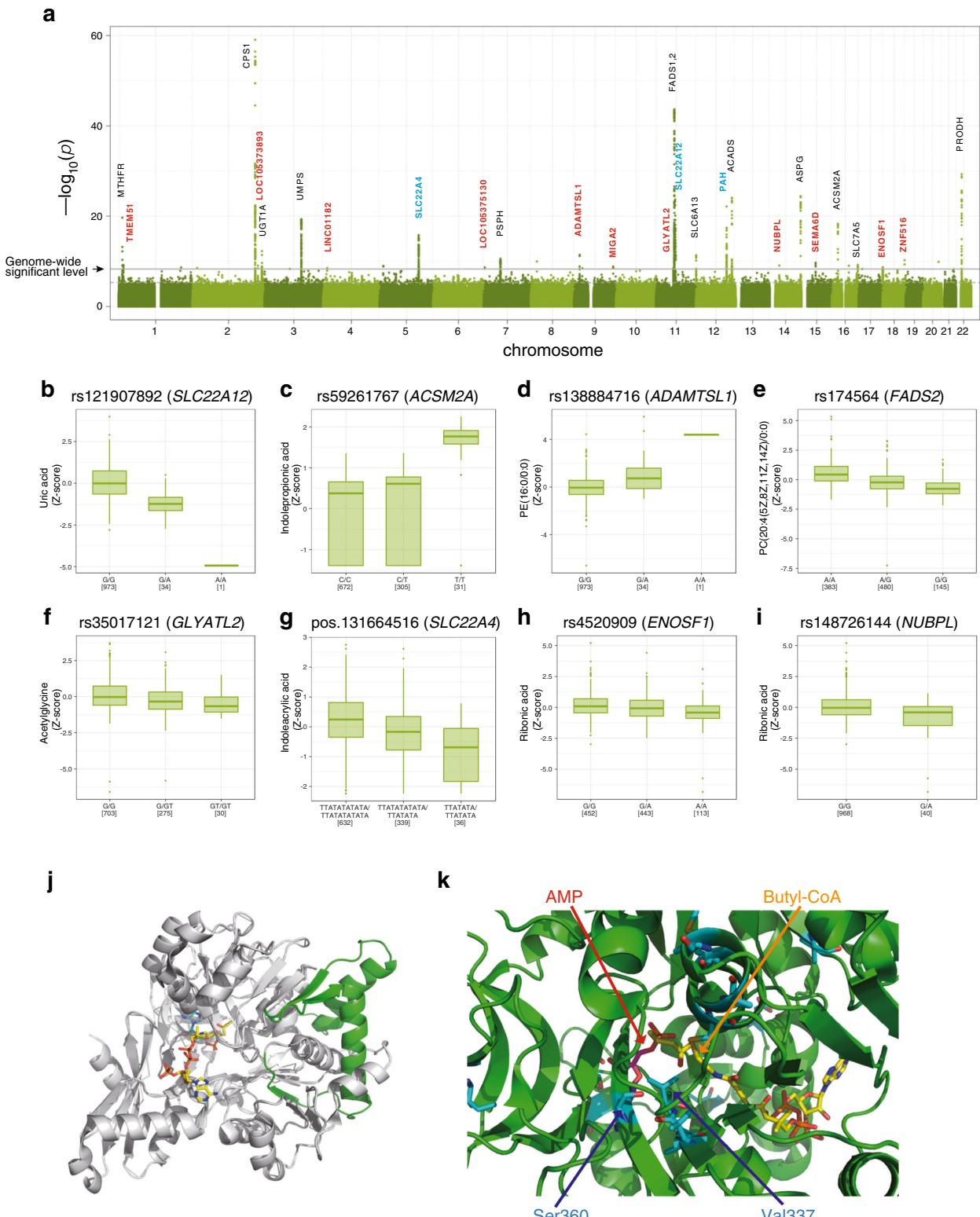

urate from the tubular lumen to the cytosol at the proximal tubules in the kidney[20]. This stop-gain variant causes the production of a truncated form of URAT1, resulting in disruption of SLC22A12 function. This variant is a causative variant for idiopathic renal hypouricemia, which causes no symptoms in many affected individuals but sometimes causes exercise-induced acute renal failure[20]. The minor allele frequency of this variant was 0.018 in the population examined in this study; this variant is

found only in East Asian ethnic groups, including Japanese and Korean populations, but not in European or African populations (Supplementary Data 3).

Another stop-gain variant is rs59261767, which is in the acyl-coenzyme A synthetase mitochondrial (ACSM2A) gene and results in a stop codon at residue 115 (R115ter) of the gene (Table 1 and Supplementary Fig. 2b). The minor allele variant of this SNP increases the concentration of indolepropionic acid in

**Fig. 1 Associations of metabolites with loci in the current metabolome genome-wide association study. a** Manhattan plots for metabolic traits. The strength of association with plasma metabolite concentrations for the 26 loci is shown based on the results from the association studies for all 1008 samples. The line indicates a suggestive genome-wide significance level with $P$-value of $4.598 \times 10^{-9}$. For gene annotations, novel associations are depicted in cyan, while novel loci are depicted in red. **b–i** Distribution of the plasma metabolites. Distributions of the plasma metabolites across the genotypes are shown using a box plot. Boxes represent the interquartile range (IQR) between the first quartile (Q1) and third quartile (Q3), and the line inside represents the median. Whiskers denote the lowest and highest values within 1.5× IQR from Q1 and Q3, respectively. Dots represent outliers beyond the whiskers. These figures were made using the R package. **j** Ribbon representation of the crystal structure of human ACSM2A (PDB ID: 3EQ6). The missed (nontranslated) region and the remaining region caused by the stop-gain variant are depicted in gray and green, respectively. **k** Structure of the ligand-binding region of human ACSM2A. Two ligands (AMP and butyl-CoA) are represented by a stick model. The residues that differ between human ACSM2A and ACSM2B are also represented by a stick model, depicted in cyan. The two residues Ser360 and Val337, which are speculated to influence the ligand specificity of the enzymes, are indicated by blue arrows.

plasma (Fig. 1c). Indolepropionic acid is a metabolite of tryptophan and is produced by symbiotic bacteria in the human gastrointestinal tract. ACSM2A is a mitochondrial enzyme that catalyzes the ligation of medium-chain fatty acids to coenzyme A, which is the first step of fatty acid metabolism[21]. This enzyme also catalyzes the detoxification of xenobiotics through glycine conjugation[22,23]. A truncated form of the ACSM2A protein due to the stop-gain variant R115ter contains neither a substrate-binding site nor catalytic residues, indicating that this variant completely abolishes the catalytic activity of ACSM2A (Fig. 1j)[21].

There are six types of acyl-coenzyme A synthetases (ACSMs) in mitochondria, and the amino acid sequence of ACSM2B is highly similar (97% identical) to that of ACSM2A, suggesting that many of the substrates of ACSM2A, especially fatty acids, may also be catalyzed by ACSM2B. However, the homozygotes of the variant allele (31 cases) exhibited a dramatic 4.7-fold increase in median indolepropionic acid concentration compared with the wild-type homozygotes (672 cases), whereas heterozygotes of the SNP (305 cases) exhibited only 1.7-fold increase, suggesting that indolepropionic acid is mainly catalyzed by ACSM2A. Indeed, the ACSM2A structure shows that residues at the substrate-binding pocket, such as V337 and S360, are substituted for other amino acids in ACSM2B (Fig. 1k)[21], indicating that there may be compounds specifically catalyzed by ACSM2A and that loss of ACSM2A function may increase the plasma concentrations of those compounds.

**Metabolite associations with synonymous variants**. We also identified 37 associations of metabolites with synonymous variants (Table 2). Of these associations, 24 associations within 16 loci were newly identified in this study, while some of the 37 associations were consistent with those reported in other studies. Importantly, four of these associations were significant only for females (Table 2). The associated variants were located mainly in or around genes encoding enzymes and transporters. We also found many variants in or around genes, which were located either in noncoding sequences or in sequences encoding proteins; we found that these genes did not encode enzymes but other type of proteins. Our results showed that many of these metabolite-change-associated genes are involved in or reported to be associated with a wide variety of common diseases, such as cardiovascular diseases, diabetes, neurological diseases, and psychiatric or cognitive disorders. We classified these associations based on the types of metabolites and have described them succinctly in the following sections.

**Associations of the synonymous variants with lipids**. We identified many synonymous variants in six novel loci associated with lipids (Table 2). Among them, genetic variants around the ADAMTSL1 gene were found to be associated with six lipids (Table 2, Fig. 1d and Supplementary Fig. 2c). The ADAMTS (a disintegrin-like and metalloproteinase with thrombospondin

type-1 motifs) superfamily includes 19 secreted metalloproteinases plus 7 ADAMTS-like (ADAMTSL) proteins that contain ADAMTS ancillary domains but lack catalytic activity[24,25]. The ADAMTS superfamily is involved in various biological processes, including connective tissue structure formation and angiogenesis. Mutations in these proteins cause human genetic disorders. While the functions of the ADAMTSL family, including those of ADAMTSL1, are not well-known, the ADAMTSL family is involved in the pathogenesis of many diseases, suggesting that this family may play important physiological roles in humans[24,25].

Of the six associated lipids, five were associated with a single SNP (rs138884716) located in the intron region of the ADAMTSL1 splice variant (upstream region of the canonical ADAMTSL1 sequence; Supplemental Fig. 3). For instance, rs138884716 was associated with an increase in plasma phosphatidyl-ethanolamine PE(16:0/0:0) levels (Fig. 1d). A missense variant in ADAMTSL1 was associated with a complex phenotype including congenital glaucoma, craniofacial, and other systemic features[26]. Because lipid and fatty acid profiles in blood are associated with glaucoma[27,28], these results support the hypothesis that ADAMTSL1 plays important roles in lipid (fatty acid) metabolism. In fact, previous GWASs and other studies reported that the ADAMTSL1 locus is associated with many kinds of phenotypes, including cholesterol (Supplementary Data 4).

Genetic variants of this gene are also associated with breast cancer prognosis[29]. Meta-analysis of stage 1 and 2 patients from four cohorts revealed that two SNPs in ADAMTSL1 are associated with early-onset disease-free survival. In addition, several genes related to ADAMTSL1, such as ADAMTS1 and ADAMTS15, are involved in initiation and progression of breast cancer[25,29]. As it has been shown that extracellular lipids play an important role in promoting breast cancer growth and progression[30], these results suggest that the changes in metabolic profiles caused by ADAMTSL1 genetic variants may also influence the prognosis of breast cancer after treatment.

Previous MGWASs reported that variants in the fatty acid desaturase (FADS) gene cluster region are associated with the concentration of many kinds of phospholipids[4,5,11]. Consistent with these reports, we confirmed that the plasma concentrations of several phospholipids were associated with genetic variants located in the FADS gene cluster region (Table 2, Fig. 1e and Supplementary Fig. 2d). Two genes in this region, namely, FADS1 and FADS2, encode rate-limiting enzymes in polyunsaturated fatty acid metabolism and are involved in a wide variety of physiological processes[31]. It has been reported that genetic variations in the FADS gene cluster region are associated with various diseases, including cardiovascular diseases, diabetes, and psychiatric diseases.

**Associations between metabolites and SNPs observed in only the female population**. In this study, we used a new reference

**Table 1 Nonsynonymous variants in genome-wide significant loci associated with metabolites.**

| Genes | Metabolites | Chr | Position (GRCh37) | rsID | Variant | Sex | p-value | MAF | gene function | Novel Locus or Association (Reference) for MGWAS studies* | Replication p-value # |
|---|---|---|---|---|---|---|---|---|---|---|---|
| CPS1 | Glycine | chr2 | 211540507 | rs1047891 | exonic (T1406N) | all | 8.711E-60 | 0.153 | amino acid metabolism | (A, B, C, D) | 3.39E-17 |
|  | Acetylglycine | chr2 | 211540507 | rs1047891 | exonic (T1406N) | all | 1.815E-22 | 0.153 | amino acid metabolism | (C) | 1.00E-05 |
| PRODH | Proline | chr22 | 18910355 | rs5747933 | exonic (T275N) | all | 4.969E-30 | 0.149 | amino acid metabolism | (A, B, C) | 2.81E-13 |
| ASPG | Asparagine | chr14 | 104571054 | rs8012505 | exonic (S344R) | all | 3.716E-25 | 0.133 | amino acid metabolism | (A, B) | 9.67E-07 |
| ACADS | Ethylmalonic acid | chr12 | 121176083 | rs1799958 | exonic (G209S) | all | 9.427E-25 | 0.111 | fatty acid metabolism | (A, C, D) | 1.91E-14 |
| SLC22A12 | Uric acid | chr11 | 64361219 | rs121907892 | exonic (stopgain W258Ter) | all | 2.091E-24 | 0.018 | transporter | Novel Association |  |
| PAH | Phenylalanine | chr12 | 103306579 | rs118092776 | exonic (R53H) | all | 7.281E-23 | 0.049 | amino acid metabolism | (A, B) | 1.36E-07 |
|  | Glutamylphenylalanine | chr12 | 103306579 | rs118092776 | exonic (R53H) | all | 3.125E-11 | 0.049 |  | Novel Association (A, B) (B) |  |
| MTHFR | Formic acid | chr1 | 11856378 | rs1801133 | exonic (A222V) | all | 2.177E-20 | 0.380 | one carbon metabolism |  |  |
| ACSM2A | Indolepropionic acid | chr16 | 20477004 | rs59261767 | exonic (stopgain R115Ter) | all | 1.123E-18 | 0.182 | fatty acid metabolism | (A, C) | 7.48E-11 |

*MGWAS study references annotation: A: Shin et al, Nat. Gen. 2014[5], B: Koshiba et al. Sci. Rep. 2016[14], C: Long et al. Nat. Gen. 2017[9], D: Yousri et al. Nat. Commun. 2018[15].
#Replication study was performed only for the associations for polymorphisms with MAF > 0.05.

panel from ToMMo (3.5KJPNv2)[17] that covers the X-chromosome. Therefore, we examined associations of metabolites and SNPs on the X-chromosome, but we could not find significant associations in the analysis.

We also investigated whether there are any sex-specific associations between metabolites and SNPs on autosomes. We examined associations between metabolites and SNPs separately in females and males and compared the score with that for the entire (female plus male) population. To our surprise, we identified five associations between metabolites and genetic variants that were restricted to only the female population. These associations include *CPS1* vs. homo-L-arginine, *PRB2* vs. SM (d18:1/18:1(9Z)), *TXNDC9* vs. 3,4,5-trimethoxycinnamic acid or trans-2,3,4-trimethoxycinnamic acid, *AGBL4* vs. phospholipid PA (20:3(8Z,11Z,14Z)/0:0), and *ZNF385D* vs. phosphatidyl inositol PI(38:4) (Table 2, Fig. 2 and Supplementary Figs. 4, 5, Notes). One of the salient examples was the association between homo-L-arginine vs. *CPS1*, which showed a strong association in females but an almost null-association in males (Fig. 2a, b). In contrast, the associations between phenylalanine vs. *PAH* and PC(42:8) vs. *FADS1,2* showed no such sex-based difference (Fig. 2k–n, respectively).

While we wished to pursue a replication study for these associations, we could not include enough number of females into the present replication study, and a solid validation for the female specific associations remains to be conducted. Similarly, the reason or mechanism for the specific accumulation of metabolite-genetic variant associations in females remains unclear at present. One plausible explanation for this phenomenon is that there may be a strong influence of confounding factors for males, leading to these female-enriched associations. Alternatively, sex hormones may influence the associations.

**Association of SNPs with glycine-related metabolites.** We also identified new loci associated with amino acids and their metabolites (Table 2). Among these compounds, acetylglycine was found to be associated with a locus containing three types of glycine N-acyltransferase genes (*GLYAT*, *GLYATL1*, and *GLYATL2*) (Supplementary Fig. 2e). These enzymes transfer an acyl group from acyl-CoA to the N-terminus of amino acids, mainly glycine (a reaction referred to as glycine conjugation)[32–34]. This reaction is important for the detoxification of xenobiotics, such as benzoate, because GLYATs target not only acyl-CoA but also xenobiotic acyl-CoA, which has the potential to sequester coenzyme A (CoASH) and inhibit several enzymes[34,35]. Among the three GLYATs, GLYAT is active for xenobiotic acyl-CoA and short/medium-chain acyl-CoAs, GLYATL2 catalyzes medium/long-chain acyl-CoAs, and GLYATL1 conjugates glutamine instead of glycine. Therefore, acetylglycine may be catalyzed mainly by GLYAT.

Our results showed that the identified minor allele variant was significantly associated with a decrease in the acetylglycine concentration in plasma, indicating that individuals with this minor allele have lower glycine conjugation activity than those with the major allele (Fig. 1f). As described above, we also identified the missense variant affecting the function of ACSM2A, another player in this detoxification system (Table 1 and Fig. 1c, j, k). Xenobiotics and some fatty acids are activated by conversion to acyl-CoA (xenobiotic-CoA) by ACSM enzymes and are then conjugated to glycine by GLYATs (Fig. 3), resulting in the excretion of glycine-conjugated xenobiotics. Importantly, this metabolic pathway heavily depends on glycine supplementation because acetylglycine is also associated with the missense variant of the *CPS1* gene, which is associated with plasma glycine levels with high significant *P*-value (8.711E–60) (Table 1). These data

**Table 2 Synonymous variants in genome-wide significant loci associated with metabolites.**

| Genes | Metabolites | Chr | Position (GRCh37) | rsID | Variant | Sex | P-value | MAF | Gene function | Novel Locus or Association (Reference) for MGWAS studies* | Replication p-value# |
|---|---|---|---|---|---|---|---|---|---|---|---|
| FADS1,2 | PC(20:4(5Z,8Z,11Z,14Z)/0:0) | chr11 | 61,588,305 | rs174564 | Intronic | All | 2.164E−44 | 0.382 | Lipid metabolism | (A, C, D, E) | 1.55E−04 |
| | PC(42:8) | chr11 | 61,597,212 | rs174570 | Intronic | All | 5.695E−33 | 0.380 | | | 7.34E−07 |
| | PC(0:0/20:4(5Z,8Z,11Z,14Z)) | chr11 | 61,571,348 | rs174548 | Intronic | All | 2.266E−18 | 0.379 | | | 3.53E−02 |
| | PC(40:6) | chr11 | 61,579,760 | rs174555 | Downstream | All | 7.003E−16 | 0.379 | | | 4.00E−02 |
| | LysoPE(20:4(5Z,8Z,11Z,14Z)/0:0) | chr11 | 61,546,592 | rs174530 | Intronic | All | 1.281E−14 | 0.386 | | | 3.61E−02 |
| | PE(18:2(9Z,12Z)/0:0) | chr11 | 61,579,760 | rs174555 | Intronic | All | 6.140E−11 | 0.379 | | | 6.14E−02 |
| | PC(42:6) | chr11 | 61,605,499 | rs174578 | Intronic | All | 1.736E−09 | 0.382 | | | 6.08E−02 |
| UMPS | Orotic acid | chr3 | 124,462,259 | rs9875527 | Intronic | All | 4.504E−20 | 0.235 | Synthase | (C) | 8.65E−03 |
| SLC22A4 | Indoleacrylic acid | chr5 | 131,664,516 (INDEL) | | Intronic | All | 1.590E−16 | 0.204 | Transporter | Novel association (A, C, D) | 8.96E−04 |
| | Phenylalanylvaline or Valylphenylalanine | chr5 | 131,659,885 | rs34376246 | Intronic | All | 4.023E−14 | 0.278 | | Novel association (A, C, D) | 5.70E−03 |
| UGT1A | Biliverdin | chr2 | 234,671,462 | rs28946889 | Intronic | All | 5.495E−13 | 0.509 | Transferase | (A, C, D) | 1.07E−04 |
| ADAMTSL1 | PE(16:0/0:0) | chr9 | 18,389,546 | rs138884716 | Intronic | All | 3.566E−12 | 0.018 | Metalloproteinase-like | Novel locus | |
| | PE(0:0/16:0) | chr9 | 18,389,546 | rs138884716 | Intronic | All | 8.478E−11 | 0.018 | | | |
| | LysoPE(18:0/0:0) | chr9 | 18,389,546 | rs138884716 | Intronic | All | 8.168E−10 | 0.018 | | | |
| | NeuAcalpha2-3Galbeta-Cer(d18:1/16:0) | chr9 | 18,389,546 | rs138884716 | Intronic | All | 1.291E−09 | 0.018 | | | |
| | PC(0:0/16:0) | chr9 | 18,389,546 | rs138884716 | Intronic | All | 2.383E−09 | 0.018 | | | |
| | PC(16:0/0:0) | chr9 | 18,337,632 | rs139140109 | Intronic | All | 4.191E−09 | 0.019 | | | |
| SLC6A13 | 3-dehydroxycarnitine or 2-amino-heptanoic acid | chr12 | 345,175 | rs20080402 | Intronic | All | 4.642E−12 | 0.407 | Transporter | (A, C) | 1.41E−04 |
| PSPH | Serine | chr7 | 56,159,209 | rs35935362 (INDEL) | Intergenic | All | 2.778E−11 | 0.649 | Amino acid metabolism | (A) | 5.00E−02 |
| ZNF516, C18orf65 | PC(0:0/20:4(5Z,8Z,11Z,14Z)) | chr18 | 74,207,594 | rs3813101 | Upstream (ZNF516), 5'UTR (C18orf65) | All | 5.502E−11 | 0.044 | | Novel locus | |
| SEMA6D | PC(20:4(5Z,8Z,11Z,14Z)/0:0) | chr18 | 74,207,594 | rs3813101 | Intronic | All | 4.725E−10 | 0.044 | | | |
| | 8-Hydroxy-5,6-octadienoic acid or 5-oxo-7-octenoic acid | chr15 | 47,769,424 | rs281320 | Intronic | All | 2.097E−10 | 0.833 | Membrane protein | Novel locus | 7.31E−01 |
| SLC7A5, TMEM51-AS1, TMEM51 | l-kynurenine | chr16 | 87,877,246 | rs58852522 | Intronic | All | 6.078E−10 | 0.114 | Transporter | (A, C) | 1.28E−01 |
| | Glutaric acid | chr1 | 15,477,364 | rs61782694 | Intronic (TMEM51-AS1), upstream (TMEM51) | All | 6.965E−10 | 0.034 | | Novel locus | |
| NUBPL | Ribonic acid | chr14 | 32,134,416 | rs148726144 | Intronic | All | 8.365E−10 | 0.020 | Nucleotide binding protein-like | Novel locus | |
| LOC105373893 | l-Palmitoylcarnitine | chr2 | 220,999,935 | rs77317066 | Intronic (ncRNA) | All | 8.744E−10 | 0.012 | | Novel locus | |
| | Oleoylcarnitine | chr2 | 220,999,935 | rs77317066 | Intronic (ncRNA) | All | 2.999E−09 | 0.012 | | Novel locus | |
| MIGA2 | l-Octanoylcarnitine | chr9 | 131,826,491 | rs566804911 (INDEL) | Intronic | All | 1.388E−09 | 0.013 | Mitochondrial fusion | Novel locus | |
| ENOSF1 | Ribonic acid | chr18 | 811,700 | rs4520909 | Intergenic | All | 1.937E−09 | 0.332 | Enolase (dehydratase) | Novel locus | 1.85E−03 |
| LOC105375130 | PC(16:1(9Z)/0:0) | chr7 | 3,143,819 | rs75355767 | Intronic (ncRNA) | All | 2.081E−09 | 0.016 | | Novel locus | |
| LINC01182 | Tryptophan | chr4 | 13,666,540 | rs35802998 | Intronic (ncRNA) | All | 2.929E−09 | 0.014 | | Novel locus | |
| GLYATL2 | Acetylglycine | chr11 | 58,608,855 | rs35017121 (INDEL) | Intronic | All | 3.814E−09 | 0.166 | Transferase | Novel locus | 1.94E−02 |
| CPS1 | Homo-l-arginine | chr2 | 211,540,507 | rs1047891 | Exonic (missense T1406N) | Female | 7.080E−16 | 0.163 | Amino acid metabolism | (C) | |
| PRB2 | SM(d18:1/18:1(9Z)) | chr12 | 11,599,279 | rs11371898 (INDEL) | Intergenic | Female | 2.527E−11 | 0.936 | Glycoprotein | Novel locus | |
| TXNDC9 | 3,4,5-Trimethoxycinnamic acid | chr2 | 99,942,848 | rs57242893 | Intronic | Female | 2.335E−09 | 0.359 | | Novel locus | |
| AGBL4 | Trans-2, 3, 4-Trimethoxycinnamic acid  PA(20:3(8Z,11Z,14Z)/0:0) | chr1 | 49,544,963 | rs117641873 | Intronic | Female | 3.344E−09 | 0.023 | Metallocarboxy-peptidase | Novel locus | |
| ZNF385D | PI(38:4) | chr3 | 21,982,634 | rs625615 | Intronic | Female | 3.769E−09 | 0.344 | | Novel association (E) | |

*MGWAS study references annotation; A: Shin et al.[5], B: Koshiba et al.[14], C: Long et al.[9], D: Yousri et al.[15], E: Tabassum et al.[16]
#Replication study was performed only for the associations for polymorphisms with MAF > 0.05 (all population only). See text.

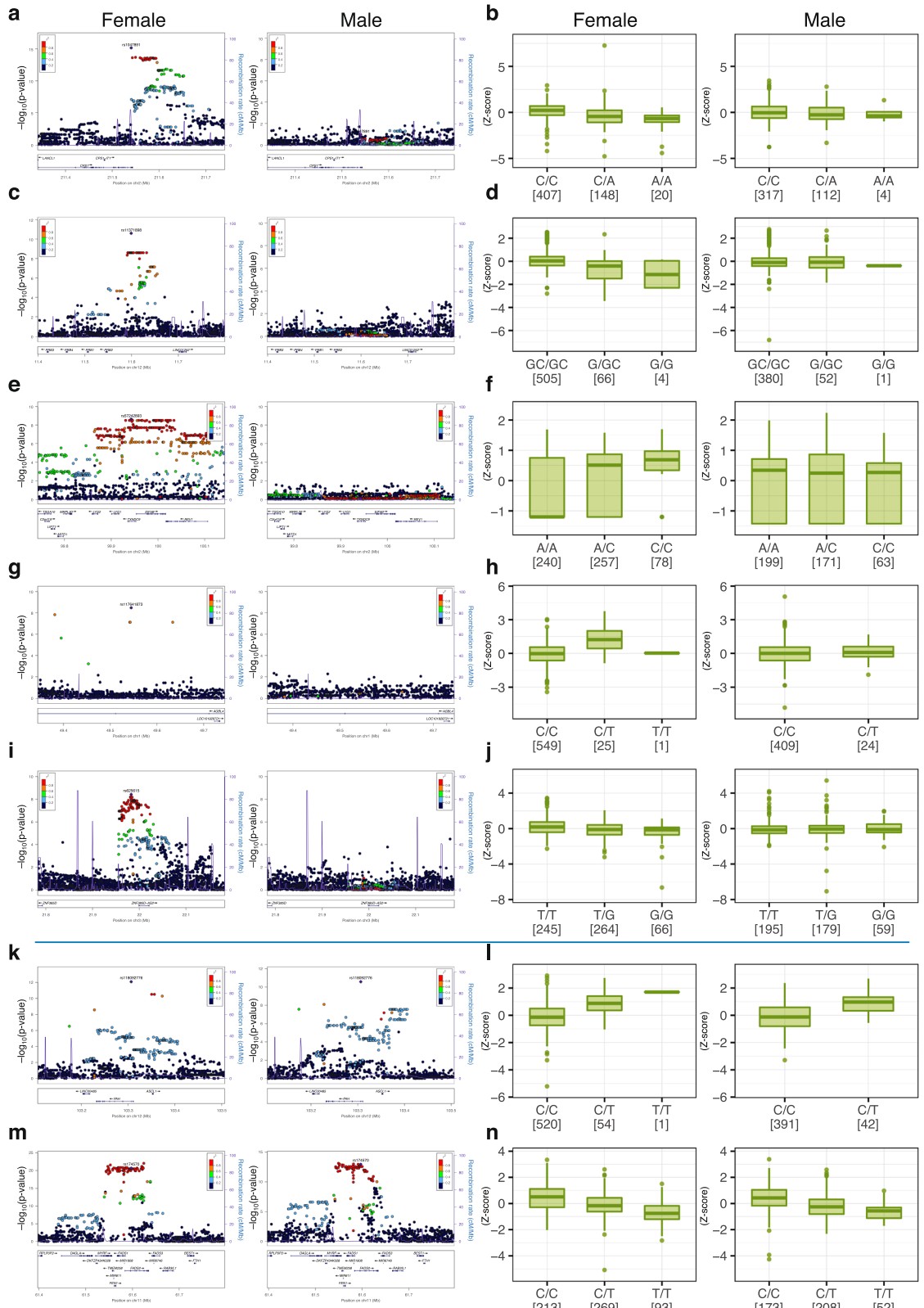

**Fig. 2 Associations significant only in the female population.** Regional association plots and box plots of five associations significant in only the female population are shown (**a–j**); **a**, **b** *CPS1* gene and homo-ʟ-arginine, **c**, **d** *PRB2* gene and SM(d18:1/18:1(9Z)), **e**, **f** *TXNDC9* gene and 3,4,5-trimethoxycinnamic acid or trans-2, 3, 4-trimethoxycinnamic acid, **g**, **h** *AGBL4* gene and PA(20:3(8Z,11Z,14Z)/0:0), and **i**, **j** *ZNF385D* gene and PI(38:4). The regional plots of the corresponding loci and the box plots of the corresponding metabolites and SNPs in male population are also shown on the right side of the female plots. For comparison, the plots and box plots of two representative associations significant for both females and males are also shown (**k–n**); **k**, **l** *PAH* gene and phenylalanine, and **m**, **n** *FADS1,2* genes and PC(42:8).

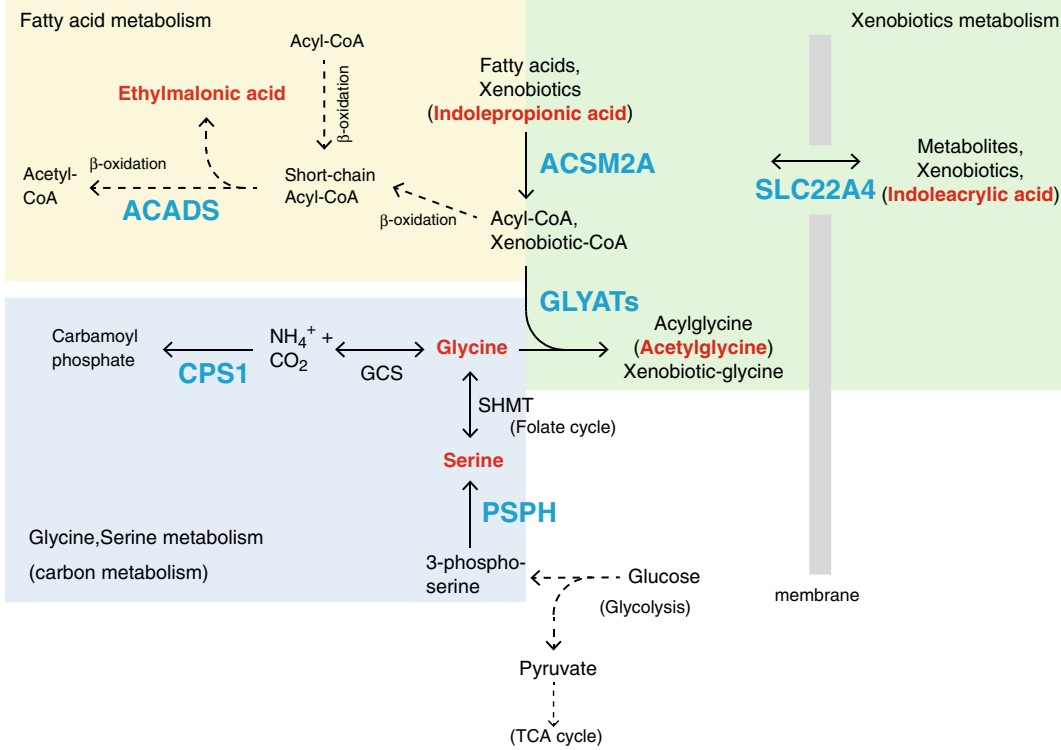

**Fig. 3 Pathways involved in glycine, fatty acid, and xenobiotic metabolism.** The associated genes and metabolites identified in this study are depicted in blue and red, respectively.

indicate that this detoxification mechanism is also affected by genetic variability in the genes involved in glycine metabolism. Previous reports shown that GLYAT expression is suppressed in hepatocellular carcinoma and GLYAT is involved in the development of musculoskeletal development[33,36]. These results support our contention that glycine conjugation plays important roles in a wide variety of biological systems.

Glycine-serine metabolism influences many kinds of metabolic pathways (Fig. 3)[37–39]. This metabolism is coupled to one carbon metabolism (folate-cycle), which is involved in de novo synthesis of nucleotides, remethylation of homocysteine (methionine-cycle), and regeneration of cofactors such as NADPH, NADH, and ATP. Glycine–serine metabolism is also involved in the production of glutathione, which regulates cellular redox balance. In addition, serine is required for the phospholipids production. On the other hand, serine (glycine) is synthesized by glycolysis from glucose or gluconeogenesis from pyruvate, and the deprivation of serine leads to reduction of glucose and glutamine metabolism (TCA cycle). These data suggest that genetic variations affecting glycine–serine metabolism also influence these metabolisms, resulting in a diversity of phenotypes.

**Associations between amino acids and SNPs in transporter genes.** We also identified four associations between metabolites, mainly amino acids and lipids, and genetic variants of transporter genes (Tables 1 and 2). Of the transporter genes, we identified that genetic variants of the *SLC22A4* gene are associated with indoleacrylic acid (Table 2, Fig. 1g and Supplementary Fig. 2f). *SLC22A4* encodes a carnitine/organic cation transporter[40] that also transports xenobiotics. Genetic variants of this transporter gene are associated with human diseases, such as rheumatoid arthritis and Crohn's disease (Supplementary Data 4)[40]. A previous report showed that indoleacrylic acid, produced from tryptophan by bacteria, promotes intestinal epithelial barrier function and mitigates inflammatory responses[41]. Although

molecular mechanism underlying these effects of indoleacrylic acid is unclear, it has been reported that treatment of peripheral blood mononuclear cells with indoleacrylic acid led to an increase in the expression of target genes in NRF2-mediated antioxidant pathways, indicating that indoleacrylic acid modifies cysteine residues of KEAP1[41].

These data indicate that perturbation of the ability to transport xenobiotics caused by genetic variations in this transporter gene may influence the susceptibility to a wide variety of human diseases. Indeed, the human gut microbiota is important for human health and that dysbiosis of the gut microbiota is associated with human diseases[42,43].

**Associations of nucleic acids and sugars with SNPs.** We also identified many associations of nucleic acids, sugars, and their metabolites with genetic variants (Tables 1 and 2). Ribonic acid, a metabolite derived from D-ribose, was associated with genetic variants of two genes, namely, enolase superfamily member 1 (*ENOSF1*) and nucleotide binding protein-like (*NUBPL*) (Fig. 1h, i and Supplementary Fig. 2g, h). ENOSF1 catalyzes the dehydration of sugars, including ribonic acid[44], and has been investigated in the cancer field because ENOSF1 expression is elevated in cell lines resistant to thymidylate synthase (TS) inhibitors, such as 5-fluorouracil (5-FU), a chemotherapeutic drug used for the treatment of many types of cancers[45]. Some patients treated with 5-FU and related drugs experienced dose-dependent toxicity, and several genetic variants of the *ENOSF1* gene were associated with toxicity[46]. These genetic variants were shown to be associated with *ENOSF1* mRNA expression but not with TS expression.

*NUBPL* encodes an iron-sulfur protein required for the assembly of the mitochondrial membrane respiratory chain NADH dehydrogenase (complex I). Mutations in the *NUBPL* gene cause mitochondrial complex I deficiency, a genetic disorder with a wide variety of symptoms[47]. While the functional relationship between NUBPL and ribonic acid is not well known,

D-ribose is essential for energy production in mitochondria, and supplementation with D-ribose, a component of the energy-producing ATP molecule, improves cellular processes under conditions of mitochondrial dysfunction[48]. These results thus indicate that ribonic acid, an oxidized form of D-ribose, may also be involved in the mitochondrial energy production process.

Consistent with a previous report[9], we also identified orotic acid, a source of uridine monophosphate (UMP), as being associated with genetic variants of uridine monophosphate synthetase (UMPS) gene. Mutations in the UMPS influence pyrimidine metabolism and result in orotic aciduria-1[49]. UMPS is involved in the conversion of 5-FU to active anticancer metabolites, and mutations in the gene contribute to 5-FU resistance in cancers[50].

**Metabolic diversity caused by combinations of common and rare variants.** While we have identified common variants that influence metabolite levels in plasma by MGWAS analysis, certain rare variants, mostly nonsynonymous, may also influence metabolite levels. To estimate how often rare variants exist and how critically rare variants influence metabolite levels, we searched for rare nonsynonymous variants or rare variants at splicing sites of the genes identified in the present MGWAS. Many of these identified genes contained certain numbers of nonsynonymous variants, mostly missense variants, while only a small number of variants were identified at splicing sites (Table 3). We investigated the effects of these rare variants on the corresponding plasma metabolite levels and found many rare variants that greatly affected the plasma metabolite levels (Fig. 4). Below, we focus on two enzymes: phenylalanine hydroxylase (PAH) and short-chain specific acyl-CoA dehydrogenase (ACADS).

PAH is the causal gene of inborn errors of metabolism, hyperphenylalaninemia (HPA) and phenylketonuria (PKU). In our MGWAS, the common variant rs118092776 (R53H) in the PAH gene is significantly associated with plasma phenylalanine levels (Table 1). Plasma phenylalanine levels of individuals with rare variants, such as rs62507335 (C265Y), were higher than the average levels of those with the heterozygous allele of the common variant R53H (Fig. 4a). Moreover, individuals with rare heterozygous alleles of the V379A, R413P, or A322T variant exhibited higher plasma phenylalanine levels than those with the common homozygous R53H alleles (Fig. 4a), indicating that some rare variants are much more effective than common variants, even in heterozygous conditions.

We mapped the common and rare variants on the structure of phenylalanine hydroxylase to elucidate the mechanism underlying these effects (Fig. 4b). The common variant R53H is located on the regulatory domain[14], while the highly effective rare variants are located on the catalytic domain or tetramerization domain. Structural analyses suggest that the C265Y substitution greatly perturbs the structure in the center of the catalytic domain, resulting in destabilization of the enzyme, while the V379A and A322T substitutions directly perturb the structure of the catalytic site, indicating considerable influences on the catalytic reaction (Supplementary Fig. 6a, b). In contrast, the R413P substitution is suggested to influence the tetramerization of the enzyme, resulting in a decrease in enzyme activity (Supplementary Fig. 6c). Thus, these rare variants markedly influence the catalytic reaction and/or the stability of the enzyme.

**Table 3 Summary of nonsynonymous rare variants and rare variants at splice sites in genome-wide significant loci identified in this study.**

| Genes | Metabolites | Number of rare variants (nonsynonymous or splicing defect) | | | Number of combination types of rare variants | Number of combination types of rare variants (over 1 S.D.) | Number of individuals with combination types of rare variants (over 1 S.D.) |
|---|---|---|---|---|---|---|---|
| | | All | Nonsynonymous (missense, nonsense, and frame-shift) | Splicing defect | | | |
| CPS1 | Glycine | 15 | 14 | 1 | 17 | 1 | 1 |
| PRODH | Proline | 24 | 23 | 1 | 34 | 13 | 19 |
| ASPG | Asparagine | 14 | 14 | 0 | 16 | 6 | 8 |
| ACADS | Ethylmalonic acid | 6 | 6 | 0 | 8 | 5 | 6 |
| SLC22A12 | Uric acid | 11 | 10 | 1 | 11 | 7 | 19 |
| PAH | Phenylalanine | 14 | 14 | 0 | 14 | 4 | 5 |
| MTHFR | Formate | 11 | 11 | 0 | 13 | 2 | 3 |
| ACSM2A | Indolepropionic acid | 16 | 16 | 0 | 19 | 7 | 7 |
| FADS1 | PC(20:4 (5Z,8Z,11Z,14Z)/0:0) | 2 | 2 | 0 | 2 | 0 | 0 |
| UMPS | Orotic acid | 10 | 10 | 0 | 12 | 4 | 20 |
| SLC22A4 | Indoleacrylic acid | 4 | 4 | 0 | 5 | 2 | 3 |
| ADAMTSL1 | PE(16:0/0:0) | 40 | 40 | 0 | 43 | 9 | 9 |
| SLC6A13 | 3-dehydroxycarnitine or 2-amino-heptanoic acid | 11 | 11 | 0 | 17 | 3 | 3 |
| PSPH | serine | 5 | 4 | 1 | 6 | 1 | 6 |
| SEMA6D | 8-Hydroxy-5,6-octadienoic acid or 5-oxo-7-octenoic acid | 14 | 14 | 0 | 15 | 4 | 4 |
| SLC7A5 | L-kynurenine | 5 | 5 | 0 | 6 | 1 | 1 |
| TMEM51 | Glutaric acid | 9 | 8 | 1 | 10 | 1 | 1 |
| NUBPL | Ribonic acid | 5 | 5 | 0 | 4 | 1 | 2 |
| MIGA2 | L-Octanoylcarnitine | 9 | 9 | 0 | 9 | 2 | 2 |
| ENOSF1 | Ribonic acid | 10 | 10 | 0 | 14 | 3 | 3 |
| GLYAT | Acetylglycine | 1 | 1 | 0 | 1 | 0 | 0 |
| ZNF516 | PC(0:0/20:4 (5Z,8Z,11Z,14Z) | 21 | 21 | 0 | 23 | 6 | 9 |

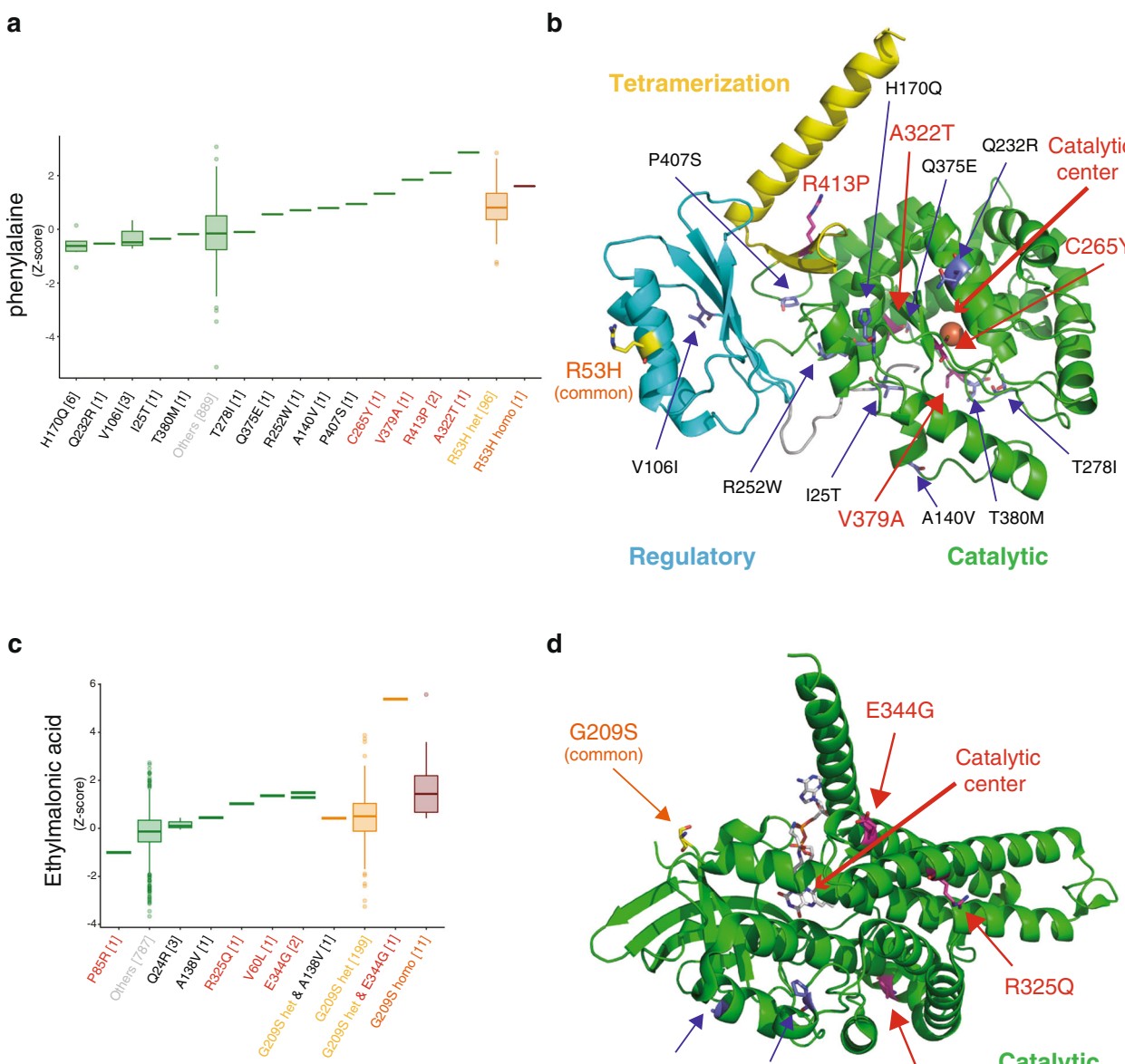

**Fig. 4 Effects of rare variants on plasma metabolite levels. a**, **c** Distributions of the plasma metabolites across the genotypes are shown using a box plot: **a** plasma phenylalanine levels with rare and common variants in the PAH gene; **c** plasma levels of ethylmalonic acid with rare and common variants in the ACADS gene. The definition of the rare variants is as follows: 1) variants around target loci, with annotation of the target gene by ANNOVAR (ver. 2017Jul16)[70]; 2) annotated as "exonic" or "splicing" in the function factor, excluding "synonymous" annotation in exonic function factor; and 3) minor allele frequency < 0.01. The box plots derived from groups with heterozygote and homozygote alleles of the common variants identified in this MGWAS study are depicted in orange and magenta, respectively, while the box plots derived from groups with no alleles of the common variants are depicted in green. The variants are labeled on the x-axis, and the type of amino acid substitution of each nonsynonymous variant is presented for each genetic group. The metabolite level was scaled as the z score (mean = 0, SD = 1), and we defined levels more than 1 SD from 0 as significant. The labels of the variants that highly influence metabolite levels are depicted in red, while those of other rare variants are depicted in black. The labels of the common variants are depicted in yellow (heterozygotes) or orange (homozygotes). The labels of samples derived from individuals with no common or nonsynonymous rare variants are depicted in gray. **b**, **d** Mapping of the rare and common variants on the structures of the enzymes. Ribbon models of (**b**) the structure of human phenylalanine hydroxylase (PAH) modeled from rat PAH (PDB ID: 5DEN) by SWISS-MODEL, and **d** the structure of human short-chain specific acyl-CoA dehydrogenase (ACADS) (PDB ID: 2VIG) are shown. The catalytic, regulatory, and tetramerization domains are depicted in green, cyan, and yellow, respectively. The residues corresponding to the position of nonsynonymous variants are shown by stick models. The cofactor and ligand in ACSDS are represented by stick models, while the $Fe^{2+}$ ion in PAH is shown by a sphere model. The catalytic side of each enzyme is indicated by an arrow.

We also found several rare variants in the *ACADS* gene that may affect plasma ethylmalonic acid levels (Fig. 4c). ACADS catalyzes the dehydrogenation step of the mitochondrial fatty acid beta-oxidation pathway[51], and impairment of ACADS activity causes short-chain acyl-CoA dehydrogenase (SCAD) deficiency, a rare autosomal recessive disorder. Consistent with a previous

report[9], our MGWAS identified a common missense variant, rs1799958 (G209S), which is associated with plasma levels of ethylmalonic acid (Table 1). We identified three rare missense variants (R325Q, V60L, and E344G) that showed much stronger effects than the common variants (Fig. 4c). Notably, we found that an individual with both the heterozygote allele of the

common variant G209S, and heterozygote allele of the rare variant E344G in combination exhibited much higher plasma ethylmalonic acid levels than those with heterozygote or homozygote alleles of the common variant G209S, indicating that the combination of the common and rare variants greatly reduced the enzyme activity (Fig. 4c).

Structural analysis showed that these three rare variants were located on the central helical region of the catalytic domain and interacted with many residues, while the common variant G209S was located at the edge of the protein, far from the catalytic side or the tetramer interface, and was exposed to the solvent (Fig. 4d). V60 is located in the core region of the protein, and substitution to leucine perturbs the local structure, while R325 interacts with E103, Q284, E322, and W329, and substitution to glutamine disrupts these interactions (Supplementary Fig. 7a, b). Finally, E344 stabilizes the N-terminus of the helix by interacting with the neighboring residues (residues 330, 334, and 340–343), indicating that substitution with glycine destabilizes the structure (Supplementary Fig. 7c). In addition, we also identified one rare missense variants (P85R) that showed stronger effect but in the opposite direction, compared to the effect of the common variant (Fig. 4c). This substitution seems to stabilize the ACADS enzyme. All these data indicate that these rare variants also perturb enzyme activity. These results support our contention that the diversity of individual metabolic phenotypes is derived from the combination of common variants with moderate effects and rare variants with much more deleterious effects.

## Discussion

We conducted extended MGWAS analyses and investigated the associations of plasma metabolites with genetic variants in Japanese populations utilizing data from 1008 participants of the TMM cohort. In addition to those reported in previous MGWASs, we have identified many novel important associations, indicating that there remain unknown associations between genetic variants and plasma metabolites. In this study, we exploited whole-genome sequence for the MGWAS, giving rise to many new associations between plasma metabolites and genetic variants. Many missense variants are associated with a wide variety of metabolites, which include substrates of the affected enzymes as well as metabolites in the pathways of these substrates, indicating that these variants moderately influence a wide range of corresponding metabolic pathways. We have proposed that the accumulation of such moderately affecting variants leads to the generation of various phenotypes in humans, including metabolic diversities and disease susceptibilities[52]. Our results support the notion that these moderately affecting variants provoke nonsynonymous changes that lead to amino acid substitutions in the marginal regions of catalytic domains or in the regulatory domains of enzymes.

We identified 37 associations between synonymous variants and metabolites. Many of these genes encode transporters or enzymes that directly interact with the associated metabolites or metabolites in related metabolic pathways. Intriguingly, however, to the best of our knowledge, some associated genes, such as AGBL4 and ZNF385D, are reported not to be involved directly in the metabolic pathways of the associated metabolites, namely, PA (20:3(8Z,11Z,14Z)/0:0) and PI(38:4). In contrast, synonymous variations of these genes are associated with unrelated plasma phospholipids and/or lipid-related diseases. In fact, SNPs in AGBL4 are associated with cardiometabolic risk and dyslipidemia[53,54]. A variant near the ZNF385D gene is associated with an increase in blood levels of Cer(42:1;2) and an increased risk of arterial and venous thrombosis[16]. It is interesting to note that SNPs in ZNF385D are associated even with language-based

learning disabilities, reading disability, and language impairment[55]. These results imply that ZNF385D may contribute to lipid metabolism in a currently unknown manner, and the MGWAS can contribute to the identification of new functions of the proteins through these findings. These observations also imply that the metabolites measured in this study are not enough to cover all associations underlying these phenotypes, and new associations of this type can be identified by increasing the number of metabolites examined in future MGWASs.

As summarized in Fig. 5, many of the metabolite-variation-associated genes identified in this study are involved in diseases, especially cardiovascular diseases and neuropsychiatric disorders, as shown in Supplementary Data 4. Indeed, genes found in this study to bear nonsynonymous variants are involved in such diseases, showing very good agreement with the findings described above and our previous report[14]. Four genes with non-synonymous variants (CPS1, MTHFR, PRODH, and PAH) are involved in inborn errors of metabolism, which lead to either accumulation or deficiency of metabolites and cause a wide range of symptoms. We also identified four different genes (NUBPL, PSPH, ACADS, and UMPS) with synonymous variants associated with metabolites. We identified 11 metabolites associated with these eight genes. Four of these metabolites are either direct substrates or products of the enzymes encoded by associated genes, but the remaining metabolites are not, indicating that variants of the genes involved in inborn errors of metabolism influence a wide range of metabolic pathways.

Many inborn errors of metabolism cause neuropsychiatric diseases, mainly because the accumulation of substrates causes toxicity in neural cells. In fact, all eight associated genes identified in this study cause symptoms of neuropsychiatric diseases. We also identified three lipid-associated genes (ADAMSTL1, ZF385D, and FADS1,2) that are involved in neuropsychiatric diseases. As summarized in Supplementary Data 4, ADAMSTL1 and ZF385D are associated with many kinds of phenotypes, including neuropsychiatric diseases and the metabolism of lipids, such as cholesterol[16,26,55–57]. Similarly, synonymous SNPs in the FADS gene cluster region are associated with dyslipidemia as well as bipolar disorder[58,59]. These results support our contention that neurological and cognitive disorders are influenced by long-term perturbations in the concentrations of metabolites.

Moreover, some of the genes (CPS1, MTHFR, and FADS1,2) that show associations with metabolites and are involved in neuropsychiatric diseases, and/or inborn errors of metabolism are also involved in cardiovascular diseases. Of these genes, CPS1 and MTHFR are involved in one carbon metabolism, while FADS1,2 are involved in lipid metabolism, suggesting that variants in these genes may influence the onset of cardiovascular diseases. These genes are also shown to be associated with many kinds of phenotypes, suggesting that genetic polymorphisms associated with cardiovascular diseases and related dyslipidemia also influence other diseases, such as neuropsychiatric diseases.

Six genes harboring metabolite-associations are involved in drug/xenobiotic metabolism. In humans, there are several metabolic pathways that eliminate xenobiotics or drugs, and half of the six genes are involved in these elimination pathways, such as glucuronidation (UGT1A) or glycine conjugation (GLYATs and ACSM2A). The other two genes (UMPS and ENOSF1) catalyze nucleotide metabolites as their native substrates, but are also involved in the metabolism of nucleotide-related drugs, such as 5-fluorouracil (Supplementary Data 4), indicating that a wide variety of metabolic pathways are involved in many kinds of drug/xenobiotic metabolism.

Our present results support the emerging notion that polymorphisms in the human genome influence the relationship between humans and gut microbes. We identified that blood

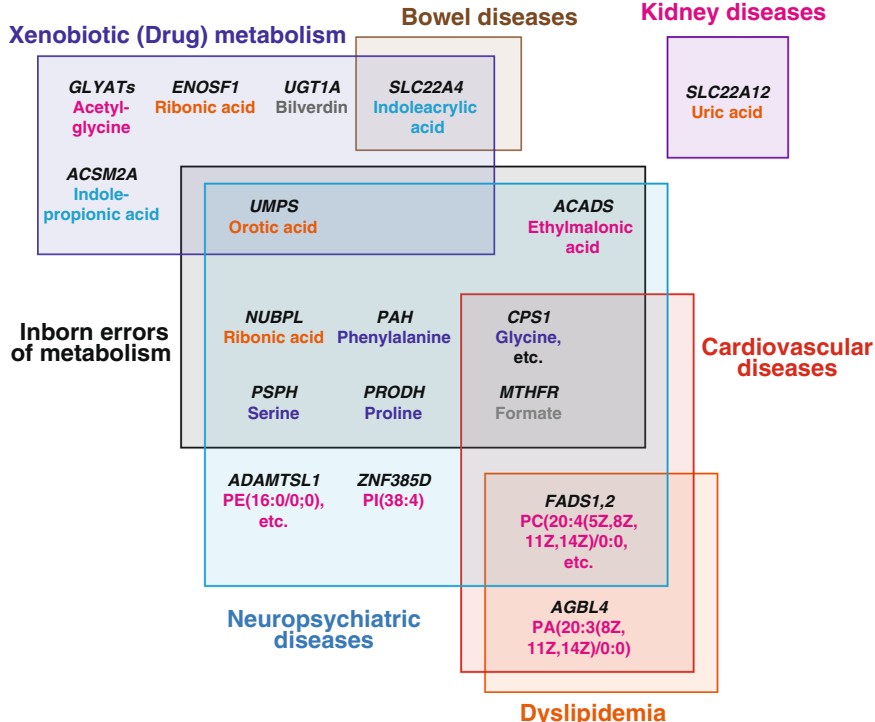

**Fig. 5 Schematic representation of the relationship of the associated genes and diseases.** The genes involved in diseases are shown on the corresponding boxes representing the disease categories. The metabolites associated with each gene are also shown, colored blue (amino acids), magenta (fatty acids/phospholipids), orange (nucleic acids), cyan (xenobiotic metabolites), and gray (others).

concentrations of two tryptophan metabolites, namely, indole propionate (IPA) and indoleacrylic acid (IA), both of which are endogenously produced by intestinal microbes, are associated with genetic variants of two human genes, namely, *ACSM2A* and *SLC22A4*, respectively. These tryptophan metabolites influence various host functions, but the directions of their contributions appear to be pleiotropic. For example, serum IPA is selectively diminished in active colitis cases compared with healthy individuals[60]. IPA and IA regulate intestinal barrier function in mice[41,61], suggesting that these tryptophan metabolites also influence intestinal barrier functions in humans. IPA also interacts with a xenobiotic sensor molecule, pregnane X receptor[61]. Similarly, IA acts as a ligand for aryl hydrocarbon receptor[41,61,62], which is involved in immune responses, indicating that IA is involved in both the gut immune system and systemic circulation. In fact, IA has both anti-inflammatory and antioxidative effects in LPS-mediated human peripheral blood mononuclear cells[41]. These diverse studies show that these tryptophan metabolites from gut microbes are closely involved in human health and diseases. Our present study indicates that polymorphisms in the human genome markedly influence the cross-talk between hosts and microbes. We propose that these genetic polymorphisms elicit active and significant effects on dietary-induced changes in human health.

To assess ethnic variations in metabolite-genome associations, we extensively compared our MGWAS results with those of previous studies: three from European populations and one from a Middle Eastern population (Tables 1, 2 and Supplementary Data 2, 3)[5,9,15,16]. We identified 30 loci (26 for both male and female and 4 only for female) in which 5 loci (*CPS1*, *ACADS*, *FADSs*, *SCL22A4*, and *UGT1A*) are observed in all three ethnic populations (i.e., Japanese, European, and Middle Eastern). By contrast, associations with nine loci (*PRODH*, *ASPG*, *PAH*, *ACSM2A*, *UMPS*, *SLC6A13*, *PSPH*, *SLC7A5*, and *ZNF385D*) are observed in Japanese and European populations, but not in the

Middle Eastern population. Among these nine loci, the *P*-values of the associations with four loci (*PAH*, *ACSM2A*, *PSPH*, and *SLC7A5*) did not reach significance in the Middle Eastern population[15], suggesting that if the number of samples increased, these associations might be also significant for the Middle Eastern population.

Of note, associations with 16 loci are observed only in the Japanese population. For some of these associations, such as those with *ADAMTSL1* and *AGBL4*, the corresponding SNPs were not observed, or their MAF were too low in the other populations (Supplementary Data 3), indicating that these associations may be specific for Japanese or East Asian populations. By contrast, 17 of 21 loci reported in the Middle Eastern populations were not detected in this study, perhaps because the corresponding SNPs were not observed in Japanese populations or the associated metabolites (or metabolite ratios) were not included in our dataset. Detailed comparisons are needed to elucidate the influence of genetic variations on metabolic phenotypes among different populations.

In conclusion, we identified a number of new associations between blood metabolites and genetic variants. Intriguingly, our analyses revealed that some of the nonsynonymous rare variants influence metabolic phenotypes much more severely than common variants identified in the corresponding genes. Furthermore, we found many metabolite-variation-associated genes that are reported to be involved in various types of human diseases, suggesting that these associated metabolites, including xenobiotics, play important functions in the pathogenesis of these diseases. With the progress of our cohort and multiomics studies, we expect to obtain more comprehensive findings regarding genome–metabolome associations.

## Methods

**Study population.** The TMM project conducts population-based prospective cohort studies with more than 150,000 participants in Japan[63,64]. The participants

in this cohort project were not selected based on any outcome or disease. For metabolome analyses, we selected 1008 adult participants (575 female and 433 male) whose whole-genome sequences had already been obtained (3.5KJPNv2 in Tohoku Medical Megabank organization: https://jmorp.megabank.tohoku.ac.jp/) [17,65]. For selection, ratio of sex and relatedness were considered, while medical history or other items in the questionnaire of the TMM cohort study were not considered. The average age of the 1008 participants was $58.9 \pm 11.5$. We calculated the relatedness of individuals and found that the pi_hat score were less than 0.125. Therefore, we concluded that there is no relatedness among the individuals who participated in this analysis. The score plots of the individuals were obtained by PCA from variants of whole-genome sequence data (Supplementary Fig. 8). These data showed that there was no population stratification in the individuals. We additionally selected 295 participants (130 female) from the same cohort for a replication study. Both the cohort study of the TMM project and the ToMMo omics study were approved by the ethics committee of Tohoku University. All adult participants signed an informed consent form.

**Metabolome analyses**. Details of sample collection and metabolome analyses are described elsewhere [14,52,66]. In brief, blood samples were collected from participants using vacutainer tubes containing EDTA-2Na [67]. Plasma samples were prepared in the ToMMo BioBank laboratory and were stored at –80 °C until metabolome analyses were performed.

For plasma metabolome analyses, we used two types of analytical methods: NMR spectroscopy and LC-MS analyses [14,66,68]. For the NMR analysis, metabolites were extracted from 200 µL of plasma samples. All NMR experiments were performed on a Bruker 600 MHz spectrometer with a SampleJet changer (Bruker BioSpin, Germany). Standard 1D NOESY and CPMG spectra were obtained from each plasma sample. The samples were analyzed using Chenomx NMR Suite 8.0 (Chenomx), and metabolites were manually quantified using the target profiling approach. Finally, we obtained the concentrations of 37 plasma metabolites from the NMR data and used these concentrations for the following MGWAS analyses (Supplementary Data 1).

We also performed a nontarget metabolome analysis with LC-MS [66]. For sample preparation, metabolites were extracted from a 50 µL plasma sample. We performed metabolome analysis using two types of LC-MS systems, depending on the nature of the metabolites. A UHPLC-QTOF/MS system (Waters) with a C18 column (Waters) was used for positive ion mode electrospray ionization (ESI), while for negative ion mode ESI, a Q Exactive Orbitrap MS system (Thermo Fisher Scientific) with a HILIC column (Sequant) was used. All data obtained from the LC-MS systems were processed by Progenesis QI data analysis software (Nonlinear Dynamics, Newcastle). We detected more than 1000 total peaks of features and identified 270 metabolites.

Details of feature selection and data processing were as reported previously [66]. First, 3200 and 5635 features were detected at the C18pos and HILICneg modes, respectively. Second, features with more than 50% missing values in all samples were removed from the data, resulting in the selection of 1444 and 3408 features. Third, the features that had both an average intensity greater than 1.0 or a detection frequency of 100% were selected, leaving 1341 and 3391 features for the C18pos and HILICneg modes, respectively. Finally, 310 and 238 features remained, respectively, after filtering on the condition that all global quality controls (gQCs) at the C18pos and HILICneg modes were detected.

We then evaluated the quality of the data set by checking the results of gQC analyses, and the batch effect of the median intensities of the gQCs was observed (Supplementary Fig. 9a). Then, the values of intensity were normalized based on the gQC values by our original software. These results showed that the any specific grouping could not be detected on the score plot of PCA after the normalization (Supplementary Fig. 9b). We finally identified the 117 and 153 features, which were manually annotated. These 270 metabolites in total were used for the subsequent MGWAS analysis (Supplementary Data 1).

To reduce the skewness and kurtosis of distribution of each metabolite or covariate, Box–Cox transformation was applied to the metabolite data and the covariate data by using the R package (car ver.2.1.5). We also performed metabolome analysis for a replication set in a similar manner to the discovery study.

**MGWAS analysis with whole-genome sequence data**. MGWAS analyses were performed for a total of 1008 samples based on both metabolome data and whole-genome sequence data [17,19,52]. A total of 1008 whole-genome sequence datasets were extracted from the whole-genome sequence data of 3552 Japanese individuals (3.5KJPNv2) from the TMM project [17]. We divided the datasets for the 1008 individuals into male and female datasets with 433 and 575 samples, respectively. We performed MGWAS for these three datasets after removing single-nucleotide variations (SNVs) with the following conditions: minor allele frequency <0.01, P-value of the Hardy–Weinberg equilibrium test <0.0001, and missing genotype rate >0.1. After filtering, the numbers of variants in the datasets with male samples, female samples, and both types of samples decreased from 28,945,113 to 11,286,983, from 32,160,803 to 11,124,783, and from 38,938,529 to 10,874,379, respectively. In the MGWAS, an additive linear regression model adjusted for BMI and age was considered, and the P-value for each variant was obtained as asymptotic P-value for t-statistic for its corresponding alleles using PLINK1.9 with

the linear option [69]. According to the Bonferroni correction, the genome-wide significance level for each dataset was set to 0.05 divided by the number of variants in the dataset, i.e., male dataset ($4.430 \times 10^{-9}$), female dataset ($4.494 \times 10^{-9}$), and the dataset for both males and females ($4.598 \times 10^{-9}$).

For replication analysis, we also performed MGWAS for 295 participants (130 female) in a similar manner to the discovery study. We selected total 24 associations for analysis because we only analyzed the significant associations of variants with an allele frequency greater than 0.05 (MAF > 0.05) for replication analysis.

**Analysis of the effect of rare variants to metabolome**. Based on the whole genome sequence datasets of a total of 1008 samples, we firstly searched for the rare variants in the genes identified in the present MGWAS. The definition of the rare variants is as follows: 1) variants around target loci, with annotation of the target gene by ANNOVAR (ver. 2017Jul16) [70]; 2) annotated as "exonic" or "splicing" in the function factor, excluding "synonymous" annotation in exonic function factor; and 3) minor allele frequency <0.01. Distributions of the plasma metabolites across the genotypes were analyzed by using the R and were described using a box plot. The metabolite level was scaled as the z-score (mean = 0, SD = 1), and we defined levels more than 1 SD from 0 as significant.

**Analysis statistics and reproducibility**. For MGWAS analyses, we used a total of 1008 samples based on both metabolome data and whole-genome sequence data [17,19,52]. The datasets for the 1008 samples were divided into male and female datasets with 433 and 575 samples, respectively. In the MGWAS, an additive linear regression model adjusted for BMI and age was considered, and the P-value for each variant was obtained as asymptotic P-value for t-statistic for its corresponding alleles using PLINK1.9 with the linear option [69]. According to the Bonferroni correction, the genome-wide significance level for each dataset was set. For replication analysis, MGWAS was performed for 295 participants (130 female) in a similar manner to the discovery study. We selected total 24 associations for analysis because we only analyzed the significant associations of variants with an allele frequency greater than 0.05 (MAF > 0.05) for replication analysis. Among a total of 24 target associations (16 loci), 13 were replicated ($P \leq 0.05/16 = 0.0031$), and 7 were nominally replicated ($P \leq 0.05$).

**Reporting summary**. Further information on research design is available in the Nature Research Reporting Summary linked to this article.

## Data availability

Summary GWAS statistics are publicly available at the Japanese Multi Omics Reference Panel website (https://jmorp.megabank.tohoku.ac.jp/). Individual genotyping results and metabolite data used for the association study are available upon request after approval of the Ethical Committee and the Materials and Information Distribution Review Committee of Tohoku Medical Megabank Organization. Source data underlying box plots shown in figures are provided in Supplementary Data 5.

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

## Acknowledgements

We thank all the volunteers who participated in TMM cohort study. We thank members of ToMMo at Tohoku University for their contribution to the establishment of the genome cohort and biobank and for their help with the metabolome analyses (http://www.megabank.tohoku.ac.jp/english/a171201). This work was supported in part by the Tohoku Medical Megabank Project from MEXT, Japan Agency for Medical Research and Development (AMED; grant numbers JP19km0105001 and JP19km0105002), Platform Program for Promotion of Genome Medicine (grant number JP19km0405001), Advanced Genome Research and Bioinformatics Study to Facilitate Medical Innovation (grant numbers JP19km0405203 and JP19km0405210), Project for Promoting Public Utilization of Advanced Research Infrastructure (MEXT), Sharing and administrative network for research equipment (MEXT), Practical Research Project for Life-Style related Diseases including Cardiovascular Diseases and Diabetes Mellitus (AMED), and Center of Innovation Program from Japan Science and Technology Agency (JST).

## Author contributions

Study design: S.K., K.K., and M.Y. Metabolomics experiments: S.K., J.I., I.N.M., Y.A., M.S., and D.S. Whole-genome sequence experiments: S.T., F.K., and G.T. Association study between metabolomics and genomics: I.N.M., G.T., and K.K. Biobank organization: N.M. Critical feedback on the manuscript: N.F. The manuscript was written by S.K., I.N.M., K.K., and M.Y. All authors reviewed the manuscript.

## Competing interests

The authors declare no competing interests.
