## [Peer Review File · Communications Biology]

Reviewers' comments:

Reviewer #1 (Remarks to the Author):

This study identifies associations between metabolite levels and WGS SNPs in a Japanese cohort. Such studies are important for understanding biological pathways in healthy and diseased individuals. With a new cohort (Japanese population) used in this study - being extended from a previous study by the authors, it is quite interesting, as previous studies focused on mainly Caucasians (except the one performed by Yousri et al, Natcomm, 2018 on a middle eastern population). Another addition is that the study uses WGS data as the study in Long et al in another new population.

The limitation of the study is that it uses a smaller set of metabolites compared to previous metabolomics GWASs which use around 1000 metabolites from an untargeted metabolomics technology (as in Long et al, NatGen 2017 and Yousri et al, NatComm 2018).

My main concerns are as follows:

The experimental design is not clear enough; Authors mentioned they use all samples and also each gender separately. How findings were validated in each of those? What are the discovery and replication cohorts?

Other details on the association analysis are missing. It has to be made clear whether there were any artifacts in the processing of samples as batch effects or others and in this case how it was handled.

Relatedness of individuals and population stratification will also affect the results and authors should show either show that such factors are not present (by computing and presenting results of the relatedness between individuals based on IBD and also plotting PCA computed from WGS variants for showing there is no population stratification).

How was the rare variant association analysis done and what defines the "significant associations" in this case. Currently, results on rare variants are expressed by vague terms as "larger" with no statistical significance.

Authors should also compare with Yousri et al (Nature communication, 2018), with whole exome common and rare variant analysis with 827 metabolites in an Asian Arab ethnicity. It is worth mentioning any compatibility of results with a closer ethnicity than European ones present on most MGWASs.

Regarding the two platforms used in the analysis, where there any overlapping metabolites between the two sets measured on the different platforms? If so it has to be mentioned. I also suggest supplementary tables with all metabolites from both platforms to be added.

Listing of all metabolites and their quality metrics should be given as a supplement with their qc results (missing values, etc) and annotations (lipids, amino acids, etc).

Qc of metabolomics data is missing. Where there any outliers? Any missing values being processed? What is the number of missing values accepted? It will be important to indicate that there are no any artifacts in the metabolomics data by plotting pca for metabolomics data and can be included as supplement as a proof that there are no artifacts.

It will be better to highlight the novel loci and novel associations in the table and manhattan plot. It is also better to highlight the functional variants in the manhattan plot - only if possible - without cluttering it, by a different color for example.

Reviewer #2 (Remarks to the Author):

The contribution by Koshiya et al. is of huge importance to the understanding of functionality of human genome and impact on environment on human health. Major challenge for this manuscript is the lack of replication in other international cohorts. The authors might consider it in the next step in more phenotype-specific studies to depict which associations are truly valid in the human race and which are driven by ethnicity.

Minor comments

1. Describe in more detail exclusion parameters for including patients into the study. Which confounders were considered?
2. For LC-MS measurements please describe procedures for batch correction and normalization.
3. Provide a table with confirmed and new hits for MGWAS.
4. Data presentation for the sections "Two newly identified stop-gain variants" and "Associations of the synonymous variants with lipids" and following chapters in results are actually a mixture of results presentation and their discussion. As the discussion does not lead to any new experiments initiated please move it to discussion section.
5. Sole association with a disease might be an artefact of overfitting. Please explain if the disease associations found reflect early or advanced stages and if these associations (or genes or chromosomal positions) have been found in other GWAS.
6. The discovery of FADS gene cluster in any GWAS is not new but rather confirmatory. You have clearly presented this in the paper.
7. Please clearly label all Y-axes in the whole manuscript. For example what are the units in Fig 1 b-i?
8. For figure 3 and data within please verify possible contribution of glycine and serine in other metabolic pathways like lipid biosynthesis, TCA or further signal transduction and discuss it.
9. Data and discussion in figure 5 need more work. At present these associations depicted are too unspecific (see my comment #5). Your data are very good please provide more stratified discussion and presentation.

Our answers to the comments of Reviewers

Reviewer #1

My main concerns

The experimental design is not clear enough; Authors mentioned they use all samples and also each gender separately. How findings were validated in each of those? What are the discovery and replication cohorts?

We thank the reviewer for the professional and helpful comments. We agree and wish to answer experimentally to the comments. To this end, we newly conducted a replication analysis during this revision period based on another set of participants of the same cohort and described results in the first section “MGWAS identified many novel genetic loci associated with plasma metabolite levels” (pp 5) and also Methods section (pp 22-24). Succinctly, we newly selected additional 295 participants (130 female) and performed MGWAS using whole-genome sequence data and metabolome data in a similar manner to the previous analysis. As the number of participants for the replication study was limited, we could analyze only significant associations of variants with more than 0.05 allele frequency ($MAF > 0.05$). As we could not include sufficient number of females to the replication study, the associations significant for only female could not be pursued. Among total 24 target associations (16 loci), 13 were replicated ($p \leq 0.05/16 = 0.0031$) and 7 were nominally replicated ($p \leq 0.05$). We have added these results to Tables 1 and 2. We would like to inform the reviewer that of the 4 remaining associations, the same association or associations of the same loci with similar metabolites were previously reported for the 3 associations. These results thus show that most of the associations found in this discovery study were replicated, and remaining ones would be replicated if the number of samples is increased.

Comment 1:

Other details on the association analysis are missing. It has to be made clear whether there were any artifacts in the processing of samples as batch effects or others and in

this case how it was handled.

We thank the reviewer for the constructive comments. We described the details on the process of feature selection in the paragraph “Metabolome analysis” of Methods (pp 23), and the variation of median intensities in all gQC before normalization and score plots of PCA after normalization were shown in Supplementary Fig. 9. All metabolites used for the final MGWAS analysis were listed in Supplementary Table 1.

Comment 2:

Relatedness of individuals and population stratification will also affect the results and authors should show either show that such factors are not present (by computing and presenting results of the relatedness between individuals based on IBD and also plotting PCA computed from WGS variants for showing there is no population stratification).

We thank for this comment. We added the description for relatedness of individuals and population stratification in the Methods section (pp 22) and showed the score plots of 1,008 individuals by PCA, computed from WGS variants, in Supplementary Fig. 8. We also calculated relatedness of individuals and found that π_{hat} is less than 0.125. Based on these results, we conclude that there is no population stratification and there is no relatedness of individuals.

Comment 3:

How was the rare variant association analysis done and what defines the “significant associations” in this case. Currently, results on rare variants are expressed by vague terms as “larger” with no statistical significance.

For the rare variant analysis, we added the definition of “significant associations” in the Figure legend of Fig. 4 (pp 32).

Comment 4:

Authors should also compare with Yousri et al (Nature communication, 2018), with whole exome common and rare variant analysis with 827 metabolites in an Asian Arab

ethnicity. It is worth mentioning any compatibility of results with a closer ethnicity than European ones present on most MGWASs.

We thank this professional advice. We compared our MGWAS results with those in Arab ethnicity (Yousri et al. 2018) and those in European ones (Shin et al. 2016, Long et al 2017, and Tabassum et al, 2018). We summarized the results in Tables 1 and 2, and Supplementary Table 2. Among the 26 identified loci in this study, associations with 5 loci (*CPS1*, *ACADS*, *FADSs*, *SCL22A4*, and *UGT1A*) were observed in all three (Japanese, European, and Middle Eastern) populations. Associations with 9 loci (*PRODH*, *ASPG*, *PAH*, *ACSM2A*, *UMPS*, *SLC6A13*, *PSPH*, *SLC7A5*, and *ZNF385D*) were only observed in Japanese and European populations, but not in Middle Eastern population. Associations with 11 loci were only observed in Japanese population. On the contrary, 17 of 21 loci reported in Middle Eastern populations were not detected in this study. We also compared the allele frequencies of the associated SNPs among three ethnicities and the results are presented in new Supplementary Table 3. We added a detailed description of the comparison among three ethnicities to the Discussion (pp 20-21).

Comment 5:

Regarding the two platforms used in the analysis, where there any overlapping metabolites between the two sets measured on the different platforms? If so it has to be mentioned. I also suggest supplementary tables with all metabolites from both platforms to be added.

We agree with this comment. Several metabolites were overlapped among C18pos mode of MS, HILICneg mode of MS, and NMR analyses. We mentioned the overlapped metabolites in Supplementary Table 1, in which those were labeled “Yes” on the column of “Overlapping”

Comment 6:

Listing of all metabolites and their quality metrics should be given as a supplement with their qc results (missing values, etc) and annotations (lipids, amino acids, etc).

We thank the reviewer for the comment. As requested, we described the quality metrics (high, middle, or low) on the column of “Quality” in Supplementary Table 1. We also described the annotation of each metabolite in Supplementary Table 1.

Comment 7:

Qcing of metabolomics data is missing. Where there any outliers? Any missing values being processed? What is the number of missing values accepted? It will be important to indicate that there are no any artifacts in the metabolomics data by plotting pca for metabolomics data and can be included as supplement as a proof that there are no artifacts.

We thank the reviewer for the professional comment. The details on the process of feature selection has been described in the paragraph “Metabolome analysis” of Methods (pp 23), and the variation of median intensities in all gQC before normalization and score plots of PCA after normalization were shown in Supplementary Figure 9, as described in the response to “Comment 1”.

Comment 8:

It will be better to highlight the novel loci and novel associations in the table and manhattan plot. It is also better to highlight the functional variants in the manhattan plot - only if possible - without cluttering it, by a different color for example.

We agree with this comment. Therefore, we have added a column for the description of novel loci/association in Tables 1 and 2, and highlighted them in the Manhattan plot of Fig. 1a by depicting gene names in red/cyan. We are sorry but could not highlight the functional variants in the Manhattan plot.

Reviewer #2

The contribution by Koshiba et al. is of huge importance to the understanding of functionality of human genome and impact on environment on human health. Major challenge for this manuscript is the lack of replication in other international cohorts.

The authors might consider it in the next step in more phenotype-specific studies to depict which associations are truly valid in the human race and which are driven by ethnicity.

We thank the reviewer for these constructive comments and we agree. Of the two main comments here, the former was also asked as *My main concerns* and the latter was asked as Comment 4 by Reviewer 1.

We wish to answer experimentally to the former comment, so that we newly conducted a replication analysis during this revision period based on another set of participants of the same cohort and described results in the first section “MGWAS identified many novel genetic loci associated with plasma metabolite levels” (pp 5) and also in Methods section (pp 22-24). Succinctly, we newly selected additional 295 participants (130 female) and performed MGWAS using whole-genome sequence data and metabolome data in a similar manner to the previous analysis. As the number of participants for the replication study was limited, we could analyze only significant associations of variants with more than 0.05 allele frequency (MAF > 0.05). As we could not include sufficient number of females to the replication study, the associations significant for only female could not be pursued. Among total 24 target associations (16 loci), 13 were replicated ($p \leq 0.05/16 = 0.0031$) and 7 were nominally replicated ($p \leq 0.05$). We have added these results to Tables 1 and 2. We would like to inform the reviewer that of the 4 remaining associations, the same association or associations of the same loci with similar metabolites were previously reported for the 3 associations. These results thus show that most of the associations found in this discovery study were replicated, and remaining ones would be replicated if the number of samples is increased.

As for the latter comment, we compared our MGWAS results with those in Arab ethnicity (Yousri et al. 2018) and those in European ones (Shin et al. 2016, Long et al 2017, and Tabassum et al, 2018). We summarized the results in Tables 1 and 2, and Supplementary Table 2. Among the 26 identified loci in this study, associations with 5 loci (*CPS1*, *ACADS*, *FADSs*, *SCL22A4*, and *UGT1A*) were observed in all three (Japanese, European, and Middle Eastern) populations. Associations with 9 loci (*PRODH*, *ASPG*, *PAH*, *ACSM2A*, *UMPS*, *SLC6A13*, *PSPH*, *SLC7A5*, and *ZNF385D*) were only observed in Japanese and European populations, but not in

Middle Eastern population. Associations with 11 loci were only observed in Japanese population. On the contrary, 17 of 21 loci reported in Middle Eastern populations were not detected in this study. We also compared the allele frequencies of the associated SNPs among three ethnicities and the results are presented in new Supplementary Table 3. We added a detailed description of the comparison among three ethnicities to the Discussion (pp 20-21).

Comment 1:

1. Describe in more detail exclusion parameters for including patients into the study. Which confounders were considered?

We thank for the professional comments. We added a detailed description of the considered confounders in the Method section (pp 22). Briefly, we considered relatedness of individuals or population stratification for the sample selection, resulting that there is no relatedness of individuals and there is no population stratification in the selected individuals. On the other hand, we did not consider medical history or other items in the questionnaire of the TMM cohort study.

Comment 2:

For LC-MS measurements please describe procedures for batch correction and normalization.

We thank the reviewer for the constructive comments. We described the details on the process of feature selection in the paragraph “Metabolome analysis” of Methods (pp 23), and the variation of median intensities in all gQC before normalization and score plots of PCA after normalization were shown in Supplementary Fig. 9. All metabolites used for the final MGWAS analysis were listed in Supplementary Table 1.

Comment 3:

Provide a table with confirmed and new hits for MGWAS.

We added a column for the description of novel loci/association in Tables 1 and 2,

and highlighted them in the Manhattan plot of Fig. 1a by depicting gene names in red/cyan.

Comment 4:

Data presentation for the sections “Two newly identified stop-gain variants” and “Associations of the synonymous variants with lipids” and following chapters in results are actually a mixture of results presentation and their discussion. As the discussion does not lead to any new experiments initiated please move it to discussion section.

We thank the reviewer for the professional comment. Accordingly, we reorganized the section; we have moved parts of the section, including the description for FADS gene cluster, to Discussion section (moved from pp 9 to pp 19). To maintain logical structures of the documents, we kept some of the descriptions in the section as they are.

Comment 5:

Sole association with a disease might be an artefact of overfitting. Please explain if the disease associations found reflect early or advanced stages and if these associations (or genes or chromosomal positions) have been found in other GWAS.

We thank the reviewer for the professional comment. We searched PheGenI database at NCBI and listed the associations reported by previous GWASs for each gene and generated a new Supplementary Table 4. We also searched OMIM database for disease annotations for each gene and further searched PubMed database to collect previous reports involved in diseases, which are also summarized in Supplementary Table 4. We believe that these surveys answer to the comment whether these associations or genes or chromosomal positions have been found in other GWAS or not. The answer is YES. Based on these examinations, we have added a detailed discussion about diseases for each gene (pp 8-10, 18-20). In addition, we added the description about stages for AMDAMTSL1 gene (pp 8-9).

Comment 6:

The discovery of FADS gene cluster in any GWAS is not new but rather confirmatory.

You have clearly presented this in the paper.

We have clearly described that the results of the associations with FADS gene cluster were not new but rather confirmatory in the Results section (pp 9).

Comment 7:

Please clearly label all Y-axes in the whole manuscript. For example what are the units in Fig 1 b-i?

We have depicted the units and/or labels for Y-axes in Fig. 1, Fig. 2, Fig 4, Supplementary Fig. 1, Supplementary Fig. 2, Supplementary Fig. 4, and Supplementary Fig. 5.

Comment 8:

For figure 3 and data within please verify possible contribution of glycine and serine in other metabolic pathways like lipid biosynthesis, TCA or further signal transduction and discuss it.

We thank the reviewer for the constructive comments. We added the description for the possible contribution of glycine and serine in other metabolic pathways to the paragraph “Association of SNPs with glycine-related metabolites” (pp 12) and Figure 3.

Comment 9:

Data and discussion in figure 5 need more work. At present these associations depicted are too unspecific (see my comment #5). Your data are very good please provide more stratified discussion and presentation.

We thank the reviewer for the professional and helpful comments. As described in the reply for Comment 5, we further investigated previous reports of diseases or associations for the genes and summarized in Supplementary Table 4. We also summarized previous reports of xenobiotic (drug) metabolism for genes in the same table. Based on these results, we specified these associations (summarized in

Supplementary Table 4) and described the stratified discussions in the Discussion section (pp 18-20).

REVIEWERS' COMMENTS:

Reviewer #1 (Remarks to the Author):

The authors have efficiently responded to the rebuttal and worked on the required amendments and added supporting results and data to make their work more convincing to the research community.

After reading the updated version of the manuscript, and the reply to the rebuttal, I suggest that the authors take the following points into consideration to make their manuscript more clear:

1-The male and female independent analysis (discovery and replication) should be clear at the start of the results (page 5). Currently, authors have amended the results to mention that females could not be replicated. A similar thing for male discovery and replication analysis should be mentioned on page 5, because later on, in subsequent sections, male and female separate analysis is mentioned (pages 9,10). It should be clear from the start if male only analysis had no findings at all and then in subsequent sections male analysis should not be mentioned at all as it causes confusion.

2-Where the X-chromosome results are mentioned (page 9) the authors indicate it didn't give any significant associations. While it is good to mention that results, yet the sentence following that confuses the reader. The reader would not know whether the following sentences describe the analysis on the X-chromosome or the other chromosomes.

Referring to that paragraph ("In this study, we used a new reference panel from ToMMo (3.5KJPNv2)¹⁷ that covers the X-chromosome for the first time. Therefore, we examined associations of metabolites and SNPs on the X-chromosome, but we could not find significant associations in the analysis (data not shown). In the analysis, we examined metabolite-genetic variant associations separately in males and females. Notably, we identified three associations between phospholipids and genetic variants that")

3-The authors found the female results could not be replicated. In this case, the sentence in the abstract or conclusion and subsequent sections in the results, (those sentences highlighting that female-specific associations were found) should be written in a more conserved way, or to reflect the fact they were not replicated. Currently, authors have highlighted the results in Table 2 for females in many of the sections in the results.

4-Authors have added some description of selecting rare variants in the legend of figure 4. However, it is important to add to the methods section, a couple of sentences explaining the rare variant selection and whether the analysis that was used is similar to the common variant analysis or burden test (or alike was used).

Reviewer #2 (Remarks to the Author):

I am very satisfied with all corrections done.

Our answers to the comments of Reviewers

Reviewer #1

The authors have efficiently responded to the rebuttal and worked on the required amendments and added supporting results and data to make their work more convincing to the research community.

After reading the updated version of the manuscript, and the reply to the rebuttal, I suggest that the authors take the following points into consideration to make their manuscript more clear:

Comment 1:

The male and female independent analysis (discovery and replication) should be clear at the start of the results (page 5). Currently, authors have amended the results to mention that females could not be replicated. A similar thing for male discovery and replication analysis should be mentioned on page 5, because later on, in subsequent sections, male and female separate analysis is mentioned (pages 9,10). It should be clear from the start if male only analysis had no findings at all and then in subsequent sections male analysis should not be mentioned at all as it causes confusion.

We thank the reviewer for the professional and constructive comments. At the end of the first section of “Results” (page 5), we have added a description about the results of male analysis.

Comment 2:

Where the X-chromosome results are mentioned (page 9) the authors indicate it didn't give any significant associations. While it is good to mention that results, yet the sentence following that confuses the reader. The reader would not know whether the following sentences describe the analysis on the X-chromosome or the other chromosomes.

Referring to that paragraph (“In this study, we used a new reference panel from ToMMo (3.5KJPNv2)17 that covers the X-chromosome for the first time. Therefore, we examined associations of metabolites and SNPs on the X-chromosome, but we could not find significant associations in the analysis (data not shown). In the analysis, we examined metabolite-genetic variant associations separately in males and females. Notably, we identified three associations between phospholipids and genetic variants that”).

We appreciate for this professional comment and we agree. Therefore, after the description of the results for the X-chromosome, we have inserted the description explaining that the following sentences describe the results of MGWAS for SNPs on other chromosomes (autosomes) (page 9).

Comment 3:

The authors found the female results could not be replicated. In this case, the sentence in the

abstract or conclusion and subsequent sections in the results, (those sentences highlighting that female-specific associations were found) should be written in a more conserved way, or to reflect the fact they were not replicated. Currently, authors have highlighted the results in Table 2 for females in many of the sections in the results.

We agree with this comment. Following the advice, we have modified the text extensively and moved a section to Supplemental information. We also deleted substantial amount of description. Succinctly, we have modified the description of Abstract in a more conserved way (page 2), and deleted the description for sex-specific associations in Introduction (page 4). We also have modified description of the female-specific associations in Results (pages 9-11) much more simple, and moved detailed description of the associations to Supplementary Notes. Finally, we added a sentence that solid validations for the female specific associations remain to be conducted.

Comment 4:

Authors have added some description of selecting rare variants in the legend of figure 4. However, it is important to add to the methods section, a couple of sentences explaining the rare variant selection and whether the analysis that was used is similar to the common variant analysis or burden test (or alike was used).

We thank the reviewer for this professional advice. We added one section describing the methods for rare variant analyses in the Methods (page 21-22).

Reviewer #2

I am very satisfied with all corrections done..

We thank the reviewer for many kind advices.